# Temporal and Spatial Variations in Carbon Flux and Their Influencing Mechanisms on the Middle Tien Shan Region Grassland Ecosystem, China

Kun Zhang [1,2,3,4], Yu Wang [2,3,5,6], Ali Mamtimin [2,3,5,6,*], Yongqiang Liu [1,4], Jiacheng Gao [2,3,5,6], Ailiyaer Aihaiti [2,3,5,6], Cong Wen [2,3,5,6], Meiqi Song [2,3,5,6], Fan Yang [2,3,5,6], Chenglong Zhou [2,3,5,6] and Wen Huo [2,3,5,6]

1   College of Geography and Remote Sensing Sciences, Xinjiang University, Urumqi 830046, China; zkun_@stu.xju.edu.cn (K.Z.); liuyq@xju.edu.cn (Y.L.)
2   Institute of Desert Meteorology, China Meteorological Administration, Urumqi 830002, China; wangyu@idm.cn (Y.W.); gaojiach@idm.cn (J.G.); ailiyaer@idm.cn (A.A.); wencong@idm.cn (C.W.); songmq@idm.cn (M.S.); yangfan@idm.cn (F.Y.); zhoucl@idm.cn (C.Z.); huowenpet@idm.cn (W.H.)
3   National Observation and Research Station of Desert Meteorology, Taklimakan Desert of Xinjiang, Urumqi 830002, China
4   Xinjiang Key Laboratory of Oasis Ecology, Xinjiang University, Urumqi 830046, China
5   Taklimakan Desert Meteorology Field Experiment Station, China Meteorological Administration, Urumqi 830002, China
6   Xinjiang Key Laboratory of Desert Meteorology and Sandstorm, Urumqi 830002, China
*   Correspondence: ali@idm.cn

**Abstract:** Grassland ecosystems are an important component of global terrestrial ecosystems and play a crucial role in the global carbon cycle. Therefore, it is important to study the carbon dioxide ($CO_2$) process in the Middle Tien Shan grassland ecosystem, which can be regarded as a typical representative of the mountain grasslands in Xinjiang. Eddy covariance (EC) and the global carbon fluxes dataset (GCFD) were utilized to continuously monitor the Middle Tien Shan grassland ecosystem in Xinjiang throughout the 2018 growing season. The findings revealed notable daily and monthly fluctuations in net ecosystem exchange (NEE), gross primary productivity (GPP), and ecosystem respiration (Reco). On a daily basis, there was net absorption of $CO_2$ during the day and net emission during the night. The grassland acted as a carbon sink from 6:00 to 18:00 and as a carbon source for the remaining hours of the day. On a monthly scale, June and July served as carbon sinks, whereas the other months acted as carbon sources. The accumulated NEE, GPP, and Reco during the growing season were $-329.49$ g C m$^{-2}$, 779.04 g C m$^{-2}$, and 449.55 g C m$^{-2}$, respectively. On the half-hourly and daily scales, soil temperature (Ts) was the main contributor to $CO_2$ fluxes and had the greatest influence on the variations in $CO_2$ fluxes. Additionally, air temperature (Ta) showed a strong correlation with $CO_2$ fluxes. The grassland ecosystems exhibited the strongest $CO_2$ uptake, reaching its peak at soil temperatures of 25 °C. Moreover, as the air temperatures rose above 15 °C, there was a gradual decrease in NEE, while $CO_2$ uptake increased. The applicability of GCFD data is good in the grassland ecosystem of the Middle Tien Shan Mountains, with correlations of 0.59, 0.81, and 0.73 for NEE, GPP, and Reco, respectively, compared to field observations. In terms of remote sensing spatial distribution, the Middle Tien Shan grassland ecosystem exhibits a carbon sink phenomenon.

**Keywords:** $CO_2$ fluxes; eddy covariance; GCFD; grassland ecosystem; Middle Tien Shan Mountains

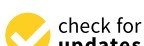



## 1. Introduction

Carbon dioxide ($CO_2$) fluxes in terrestrial ecosystems play a crucial role in the study of the global carbon cycle and carbon balance. Changes in land–air carbon exchange can cause significant fluctuations in $CO_2$ concentrations, which have important implications for global climate change [1,2]. Grassland ecosystems, as an important component of global terrestrial

ecosystems, play a crucial role in the carbon cycle of global terrestrial ecosystems [3,4]. Among all of the terrestrial ecosystems worldwide, grassland ecosystems have the second highest carbon stock after forest ecosystems, reaching 308 Pg C. Simultaneously, grasslands play a critical role in maintaining both the global carbon cycle and the structural stability of ecosystems [5]. Many scholars have explored the carbon sequestration capacity of grassland ecosystems through various methods such as eddy correlation techniques [6], experiments on the control of temperature and precipitation [7,8], as well as the coupling of remote sensing and models [9]. However, at present, most of the research on $CO_2$ fluxes in grassland ecosystems in China is focused on desert grasslands, typical grasslands, and alpine grasslands [10–12]. There is uncertainty in accurately estimating carbon flux in mountain grasslands.

Xinjiang has abundant grassland resources, covering a total area of 572,590 km$^2$ and a net area of 480,070 km$^2$, accounting for 34.68% of Xinjiang's total area and 14.50% of China's total area. Furthermore, the natural grasslands in Xinjiang are mainly distributed in mountainous areas, which account for 58% of Xinjiang's total grassland area and are the primary location for Xinjiang's grassland animal husbandry operations [13]. Simultaneously, the diversity of grassland ecosystems, types, and species creates favorable conditions for mitigating the natural shortcomings of grasslands and also provides opportunities for the development of efficient livestock industries [14]. Global climate change and regional human activities have significantly influenced the carbon cycling processes in the Middle Tien Shan grassland ecosystem [15]. Currently, research on carbon fluxes in the Tien Shan region of Xinjiang mainly focuses on studying soil carbon stocks in grasslands on the northern slopes of the Tien Shan Mountains, with a lack of studies on ecosystem carbon fluxes using the eddy covariance (EC) system [16]. Moreover, the grassland vegetation in the Tien Shan Mountains of Xinjiang exhibits distinct vertical and horizontal zonal distribution patterns. Furthermore, various types of grasslands can be found in the Tien Shan Mountains, including mountain desert, mountain desert grassland, mountain grassland, mountain meadow grassland, mountain meadow and alpine meadow grassland. Meanwhile, these representative grassland types play a crucial role in maintaining ecological balance and protecting biodiversity, serving as natural green barriers for the sustainable utilization of desert oases. The EC system is a technique that can monitor carbon fluxes over an extended period and is widely used in various ecosystem areas [17–19]. Therefore, utilizing the EC system to measure changes in carbon fluxes in the Middle Tien Shan grassland ecosystem and to investigate their relationship with environmental factors not only addresses the deficiency of carbon flux data in this ecosystem but also provides a scientific basis for studying the carbon cycle in grassland ecosystems throughout the region and even in China.

Remote sensing technology enables the transition from site-specific research to regional scales. The development of satellite data for carbon monitoring has provided possibilities for this research. GOSAT and OCO, as the main greenhouse gas satellites, are widely used in the study of carbon flux spatiotemporal variations [20–23]. With the development and integration between disciplines, carbon flux estimation methods based on machine learning, satellite remote sensing data, and reanalysis data have been developed. Simultaneously, large-scale land ecosystem model estimations have benefited from the development of global flux sites, such as Ameri Flux, Euro Flux, and China Flux. These flux towers have also promoted the study of carbon cycling in terrestrial ecosystems [24]. In recent years, the development of data fusion techniques combining global flux site data, reanalysis data, and remote sensing data has provided support for the fine-scale representation of carbon flux on remote sensing scales [25,26].

Although research using EC systems and remote sensing data to study the carbon flux and carbon cycling processes in grasslands is increasing, research on carbon flux in the alpine grassland ecosystem of the Tien Shan Mountains is currently limited. In previous studies, most research focused on either using eddy covariance data or remote sensing data separately to study carbon flux [27,28]; there is a lack of research that combines both approaches. Therefore, the objectives of our study are (1) to investigate the temporal

variations in carbon flux in the alpine grassland ecosystem of the Tien Shan Mountains, (2) to explore the main influencing factors of carbon flux in relation to meteorological factors, and (3) to present a remote sensing spatial distribution map of carbon flux in the Tien Shan region and analyze the spatial distribution pattern of carbon flux in the alpine grassland ecosystem of the Tien Shan Mountains. The hypotheses of our study are as follows: (1) the alpine grassland ecosystem of the Tien Shan Mountains has a strong carbon assimilation capacity during the daytime and peak growing season, (2) temperature has a significant influence on carbon flux absorption and emission, and (3) from the perspective of remote sensing spatial distribution, the alpine grassland region of the Tien Shan Mountains acts as a carbon sink during the growing season. This study focuses on revealing the spatial and temporal patterns of carbon flux in the Central Tian Shan grassland ecosystem and the influencing mechanisms. It provides a reference for the study of carbon flux in grassland ecosystems in other regions and theoretical support for the study of carbon balance and carbon mechanisms in other terrestrial ecosystems.

## 2. Materials and Methods

### 2.1. Study Site

The Tien Shan Mountains are located in the hinterland of the Eurasian continent, spanning the entire territory of the Xinjiang Uygur Autonomous Region (Figure 1a). With a total length of about 2500 km and a width of about 250–350 km from north to south, it is the world's largest independent latitudinal mountain system and also the farthest from the ocean [29]. The region has a typical temperate continental climate with severe winters and hot summers, a large annual temperature difference, an average annual temperature of 7.26 °C, and average annual precipitation of 257.61 mm [30]. Due to the influence of westerly circulation and the interplay of high Arctic air masses and warm and humid air currents from the Indian Ocean, the regional temperature and humidity in the area are highly variable. Ulastai station is situated in the central area of the Tien Shan Mountains (43°28′55.88″N, 87°12′5.76″E) at an altitude of 2036 m (Figure 1c) and its unique topography makes pasture the dominant grassland type in the area [31] (Figure 1b).

The footprint source area was analyzed using the Kljun footprint model [32] (Figure 2). In the range of 90% of flux footprint, portions of the footprint occupied by mountain grassland from April to September were 34%, 55%, 58%, 57%, 56%, and 55%, respectively. The radius of the 90% flux footprint fell within 1000 m, which indicated that the measured flux was majorly sourced from the mountain grassland surface and that the measured fluxes are representatives of this mountain grassland.

### 2.2. Data Source

#### 2.2.1. Measured Data Source

In 2016, the Urumqi Institute of Desert Meteorology, part of the China Meteorological Administration, established the Middle Tien Shan Grassland Land–Air Interaction Observation and Experimental Station in the Ulastai area of Baiyanggou, Urumqi. The station is equipped with an eddy covariance system, a radiation observation system, and a gradient detection system (Figure 3a). The data for this study were mainly obtained from three systems: (I) The eddy covariance system, which consists of a 3D sonic anemometer (CSAT3, Campbell Scientific, Logan, UT, USA) and an open-path infrared gas analyzer (Li7500A, Licor, Lincoln, NE, USA) (Figure 3c); we measure carbon flux using these two instruments. (II) The radiation observation system, which includes a net radiation sensor (CNR01, Kipp&Zonen, Logan, UT, USA, Figure 3b), a soil moisture sensor (CS616, USA), and a heat flow plate (HFP01, The Netherlands), was used to measure long-and short-wave radiation, soil moisture, and soil heat flux at different depths, respectively. (III) The gradient observation system includes a temperature and humidity sensor (HMP45C, Finland) for monitoring meteorological elements, such as soil temperatures at depths of 0 cm, 5 cm, 10 cm, and 20 cm, as well as air temperatures at heights of 2 m and 10 m. Figure 3d displays a 2D anemometer used to measure wind speed and wind direction. It is installed at a height

of 2 m above the ground. It is worth noting that the underlying surface of the EC station is primarily grass, with a relatively dense coverage and good representativeness (Figure 3e).

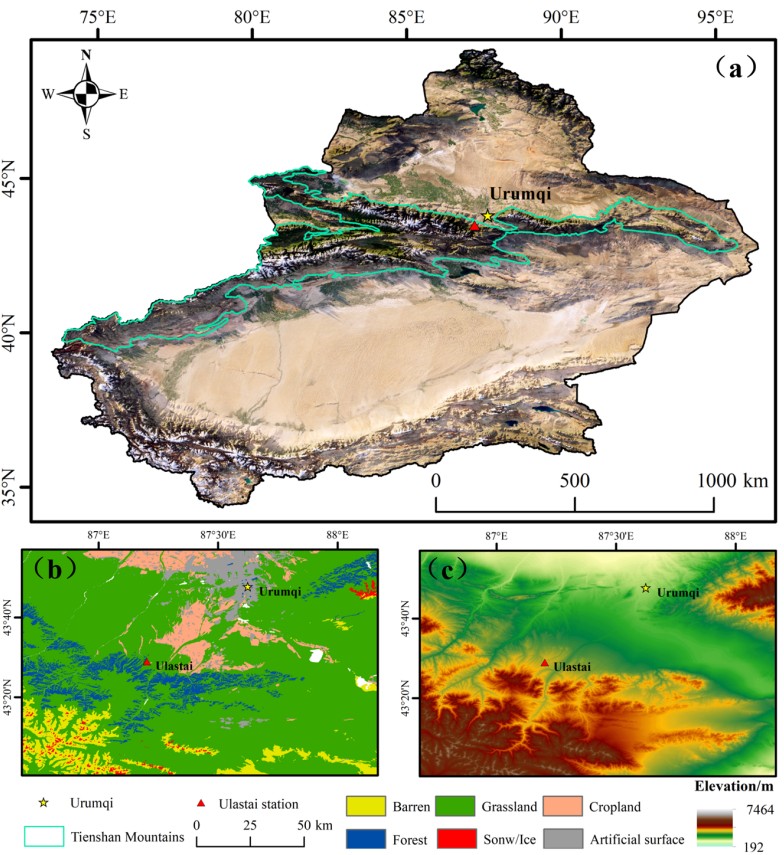

**Figure 1.** (**a**) The specific locations of the Tien Shan and Ulastai station, (**b**) the land use types at the Ulastai station, and (**c**) the elevation map of the Ulastai station.

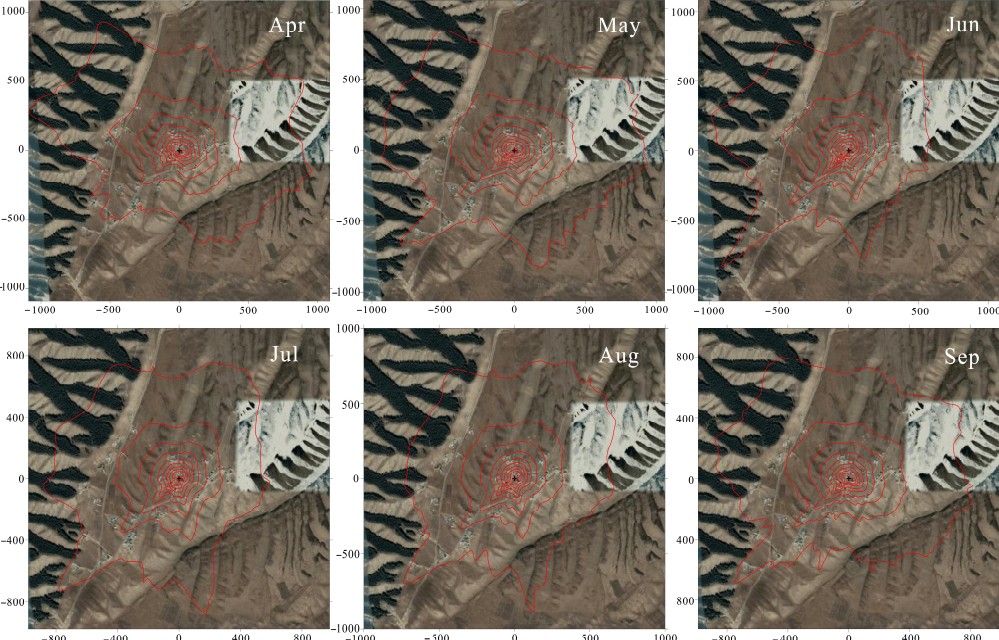

**Figure 2.** The flux footprints of the test point from April to September of 2018. The outlines enclose 10–90% of the flux footprints.

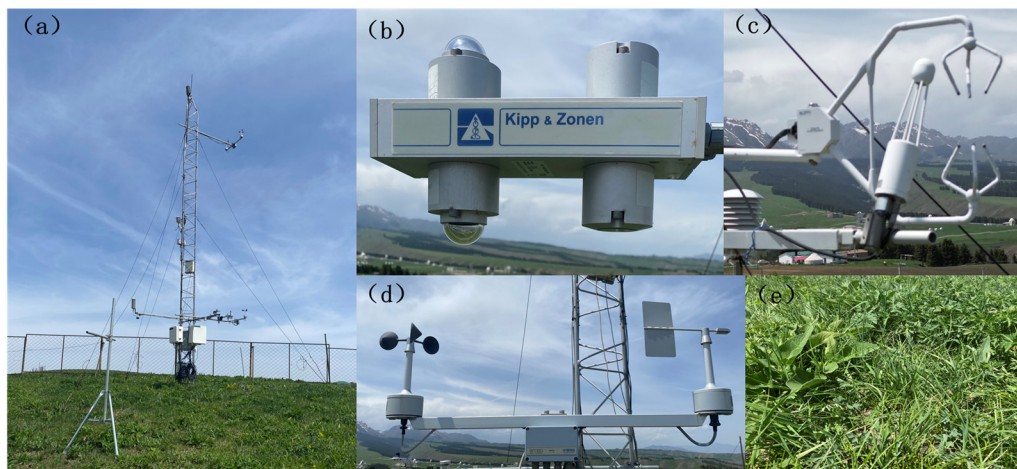

**Figure 3.** (**a**) Ulastai station flux tower, (**b**) radiation observation system, (**c**) eddy covariance system; (**d**) 2D anemometer, and (**e**) photo of the underlying grassland, taken on 8 May 2023.

The measured data were output through a data collector (CR3000, Logan, UT, USA) at time intervals of 10 s, 1 min, 30 min, and 1 h. The radiation observation system and the gradient observation system were acquired at a frequency of 1 Hz, while the eddy covariance system was acquired at a frequency of 10 Hz. The time scale of this study is from April to September 2018, and all times used are local time.

### 2.2.2. Remote Sensing Data

The global carbon fluxes dataset (GCFD) is based on field carbon flux data, meteorological data, and remote sensing data. It utilizes convolutional neural network models (CNNs), artificial neural network models (ANNs), and random forest (RF) models to generate a global carbon flux dataset that includes net ecosystem exchange (NEE), gross primary productivity (GPP), and ecosystem respiration (Reco). The dataset has been validated and analyzed, revealing that NEE has lower accuracy compared to GPP and Reco in terms of temporal variability. However, the GCFD dataset fills in the gaps caused by the uneven distribution of global flux sites and the shortage of data in certain regions. The dataset has a spatial resolution of 1 km at three time steps per month from January 1999 to June 2020. The GCFD can be a useful reference for various meteorological and ecological analyses and modelling, especially when high resolution carbon flux maps are required [33] (Table 1). As per the GCFD data, the time scale is from April to September 2018, with a step of every 10 days, totaling 18 remote sensing images. We downloaded the GCFD dataset through the National Tibetan Plateau Data Center (https://dx.doi.org/10.11888/Terre.tpdc.300009, (accessed on 1 June 2023)).

**Table 1.** Detailed information about the GCFD dataset.

| Data Parameters | Detailed Information |
| --- | --- |
| Variable | NEE, GPP, Reco |
| Time coverage | 1990–2020 |
| Spatial resolution | 1 km |
| Coverage area | 180°E~180°W, 80°N~60°S |
| Data storage format | NetCDF4 |
| Data units | $0.01 \text{ gc m}^{-2} \text{ d}^{-1}$ |
| Missing values | 0 |

### 2.3. Data Processing

2.3.1. Measured Data Processing

The EC system continuously observes $CO_2$ flux, and the 30 min data observed via the EC system is processed using the EddyPro 7.1 software to ensure the accuracy of the observations. The original data underwent several processes to ensure accuracy. Initially, any outliers were removed. Subsequently, trend correction [34], the double rotation method [35], frequency response correction [36], sonic temperature correction [37], and Webb–Pearman–Leuning correction [38] were applied. Additionally, a turbulence stationarity test and an overall turbulence characteristic test were conducted to assess the quality of the data. Any data marked with flag 2 were excluded, and data with a frictional wind speed below the nighttime threshold were also eliminated.

Data imputation is particularly important due to various reasons, such as weather conditions, instrument failure, and quality control, which led to 22% of the $CO_2$ flux data being missing from April to September of 2018. This issue causes inconvenience in the application of flux tower data. In the current investigation, an online data interpolation tool developed by the Biochemistry Department of the Max Planck Institute (https://www.bgc-jena.mpg.de/bgi/index.php/Services/REddyProcWeb, (accessed on 15 April 2023)) was used to interpolate the $CO_2$ flux data. The method employs REddyProc in the R language package for data interpolation, which relies on meteorological data: total radiation, air temperature, and saturated water vapor pressure deficit (VPD) [39]. The data interpolation process consists of the following four steps: (I) Detection and rejection of (NEE) outliers. (II) Estimation of the night-time friction wind speed u* threshold. When the atmospheric turbulent motion is weak, the friction wind speed decreases, and the NEE measured via the eddy covariance system underestimates; therefore, the NEE data below the night-time friction wind speed u* threshold needs to be removed [40,41]. (III) Interpolation of NEE data by screening out data with time gaps and interpolating NEE using relevant meteorological and flux data [39]. (IV) The night-time data splitting method assumes that Reco is only related to temperature changes and that night-time vegetation only undergoes respiration. Therefore, the night-time NEE response curve to temperature can be used to infer the changes in vegetation *Reco* during the daytime. Finally, GPP can be calculated using Formula (1). The daytime data splitting method for *NEE* assumes that the relationship between daytime NEE and total radiation (*Rg*) and vapor pressure deficit (VPD) affects *GPP*, as well as the effect of temperature on Reco.

$$NEE = Reco - GPP \tag{1}$$

Due to the lack of instrumentation to observe saturated water vapor pressure, the empirical Tetens Equation (2) was used to indirectly estimate the saturated water vapor pressure from the air temperature using the relationship between the saturated water vapor pressure and the air temperature, as follows [42],

$$e^0(T) = 0.6108 \exp\left[\frac{17.27T}{T + 237.3}\right] \tag{2}$$

where $T$ (°C) is the air temperature and $e^0(T)$ (ka) is the saturated water vapor pressure at temperature T.

2.3.2. Remote Sensing Data Processing

The GCFD data primarily utilize remote sensing and meteorological data as predictive factors. Remote sensing variables include the fraction of absorbed photosynthetically active radiation (FAPAR) and leaf area index (LAI), while meteorological variables include 2 m temperature (Ta), surface solar radiation downward (SW_IN), latent heat flux (LE), and sensible heat flux (H). FLUXNET 2015, FLUXNET-CH4, and Drought-2018 were used as site datasets, which include meteorological data and carbon flux variables. The remote sensing data, including FAPAR and LAI, were sourced from the Copernicus Global Land

Service (CGLS) version 2 dataset, which has a 1 km resolution and covers the period from January 1999 to June 2020. The meteorological variable reanalysis data were sourced from the EAR5-Land dataset. With the above data, machine learning models are employed to predict global-scale NEE, GPP, and Reco.

Therefore, using GCFD data, the corresponding data for the Ulastai station is extracted and compared with the field-measured data for validation. The suitability of the GCFD data in the Ulastai region was evaluated using Pearson correlation coefficient (*R*), root mean square errors (*RMSEs*), and *Bias* indicators.

$$R = \frac{\sum_{i=1}^{n}(x_i - \overline{x})(y_i - \overline{y})}{\sqrt{\sum_{i=1}^{n}(x_i - \overline{x})^2}\sqrt{\sum_{i=1}^{n}(y_i - \overline{y})^2}} \tag{3}$$

$$RMSE = \sqrt{\frac{1}{n}\sum_{i=1}^{n}(x_{o,i} - x_{m,i})^2} \tag{4}$$

$$Bias = \frac{1}{n}\sum_{i=1}^{n}(x_{o,i} - x_{m,i}) \tag{5}$$

In the equation, $x_i$ represents the measured values and $y_i$ represents the GCFD data. $\overline{x}$ and $\overline{y}$ are the average values of the GCFD data and measured values, respectively; and $x_{o,i}$ and $x_{m,i}$ represent the measured values and GCFD data at each time step, respectively.

The specific methodological flow is shown in Figure 4.

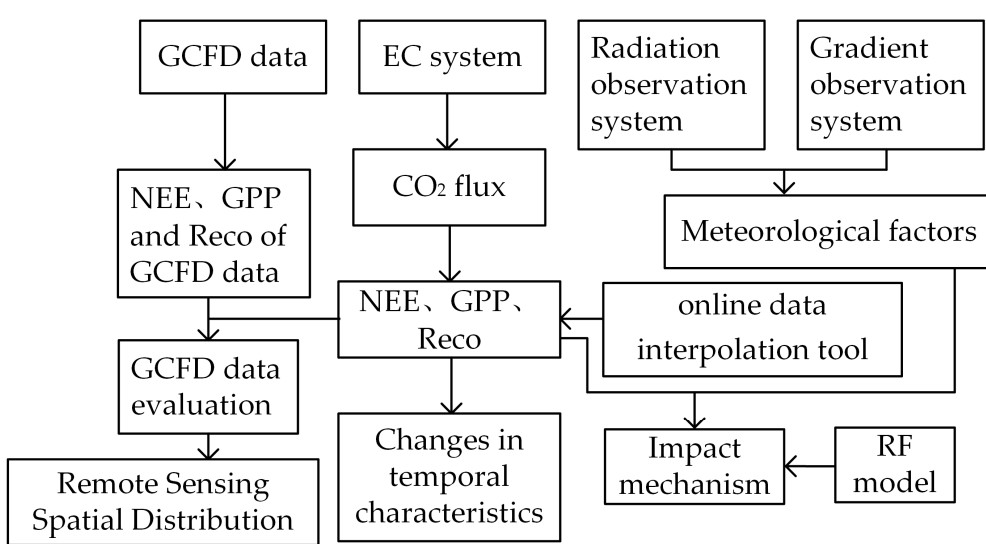

**Figure 4.** Flowchart of the methodology in this study.

## 3. Results

### 3.1. Variation in $CO_2$ Fluxes

3.1.1. Diurnal Variation in $CO_2$ Fluxes

The results from the current investigation indicated the diurnal variations in $CO_2$ fluxes during the growing season in the Middle Tien Shan grassland from April to September of 2018. NEE and GPP showed a "U"-shaped and inverted "U"-shaped trend, respectively, while Reco remained relatively constant with a mean value of 0.1 mg m$^{-2}$ s$^{-1}$ (Figure 5a). At 5:00, NEE showed a decreasing trend, while GPP showed an increasing trend. As the absorption rate of $CO_2$ increased, the concentration of $CO_2$ in the air gradually decreased, eventually causing NEE to reach a minimum value of $-0.37$ mg $^{-2}$ s$^{-1}$ at 11:00. After that, GPP gradually decreased as solar radiation weakened and photosynthesis in the grassland ecosystem slowed down, causing Reco to become the main source of $CO_2$ fluxes at night and the concentration of $CO_2$ in the air to rise. Overall, from 6:00 to 18:00,

the grassland ecosystem acted as a carbon sink, while for the rest of the time, it acted as a carbon source.

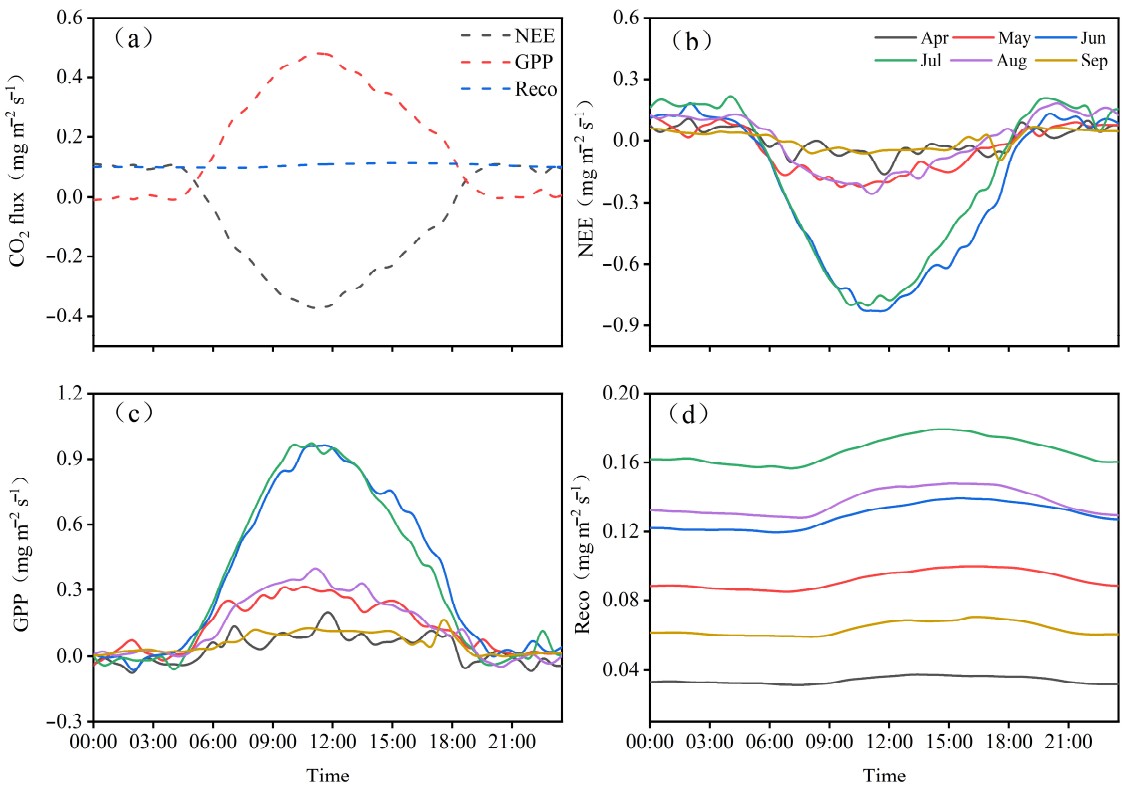

**Figure 5.** (**a**) Diurnal variation in $CO_2$ flux from April to September of 2018, (**b**) diurnal variation in net ecosystem exchange (NEE) from April to September of 2018, (**c**) diurnal variation in gross primary productivity (GPP) from April to September of 2018, and (**d**) diurnal variation in ecosystem respiration (Reco) from April to September of 2018.

NEE, GPP, and Reco varied in each month of the growing season (Figure 5b–d). As solar height and solar radiation are stronger from June to August than in other months, the changes in NEE and GPP were most pronounced in June and July, while the changes in Reco were most pronounced in July and August. Thus, carbon sources and sinks in grassland ecosystems showed greater changes in June and July than in other months. The daily average values of NEE ranged from $-0.24$ to $0.004$ mg m$^{-2}$ s$^{-1}$, GPP ranged from $0.03$ to $0.37$ mg m$^{-2}$ s$^{-1}$, and Reco ranged from $0.03$ to $0.17$ mg m$^{-2}$ s$^{-1}$ from April to September of 2018.

### 3.1.2. Daily and Monthly Variations of $CO_2$ Fluxes

Figure 5 shows the daily and monthly variations in $CO_2$ fluxes during the growing season in the Middle Tien Shan grassland from April to September of 2018. NEE, GPP, and Reco all exhibit significant seasonal variation, with NEE and GPP showing opposite trends (Figure 6a). The maximum $CO_2$ uptake occurred on 17 June, at $-9.73$ g C m$^{-2}$ d$^{-1}$, and NEE showed $CO_2$ uptake from 15 May to 6 August, during which time the grassland ecosystem acted as a carbon sink. GPP varied from $-2.48$ to $13.18$ g C$^{-2}$ d$^{-1}$, with a mean value of $4.26$ g C m$^{-2}$ d$^{-1}$. Reco variation was relatively stable, ranging from $0.12$ to $5.63$ g C m$^{-2}$ d$^{-1}$, with a mean value of $2.46$ g C m$^{-2}$ d$^{-1}$, which was significantly lower than GPP. In addition, Reco showed a slow upward trend in June and July, due to increased solar radiation and vigorous grass growth compared to other months, resulting in enhanced ecosystem respiration.

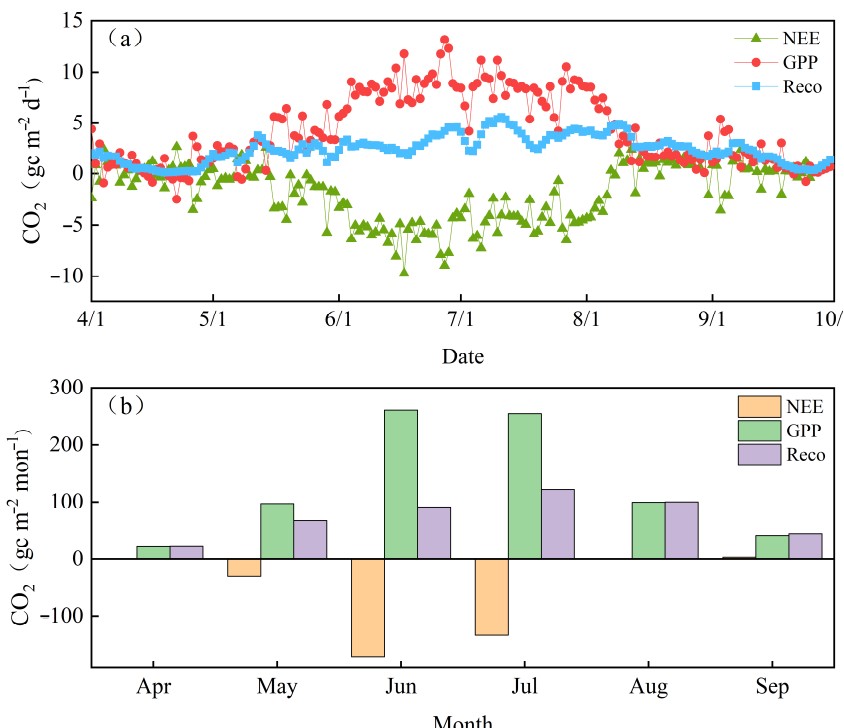

**Figure 6.** (**a**) Daily variation in net ecosystem exchange (NEE), gross primary productivity (GPP), and ecosystem respiration (Reco) in grassland ecosystems from April to September of 2018, (**b**) monthly variations of net ecosystem exchange (NEE), gross primary productivity (GPP) and ecosystem respiration (Reco) in grassland ecosystems from April to September of 2018.

The monthly variation in $CO_2$ fluxes in grassland ecosystems is shown in Figure 6b. The maximum monthly accumulation of NEE and GPP occurred in June, with $-170.01 \text{ g C m}^{-2} \text{ mon}^{-1}$ and $261.27 \text{ g C m}^{-2} \text{ mon}^{-1}$, respectively, and the maximum monthly accumulation of Reco occurred in July, with $121.80 \text{ g C m}^{-2} \text{ mon}^{-1}$. During June and July, grassland ecosystems were in their peak growing season, showing $CO_2$ flux uptake, which was manifested as carbon sinks, while the rest of the months acted as carbon sources. April and September, the beginning and end of the growing season, respectively, were the months when photosynthesis and respiration were weaker in grass compared to other months. Therefore, the uptake or release of $CO_2$ fluxes was lowest throughout the growing season. The overall accumulation of NEE, GPP, and Reco during the growing season was $-329.49 \text{ g C m}^{-2}$, $779.04 \text{ g C m}^{-2}$, and $449.55 \text{ g C m}^{-2}$, respectively.

*3.2. Impact of Meteorological Factors on $CO_2$ Fluxes*

3.2.1. Seasonal Variation in Meteorological Factors

Figure 6 illustrates the changes in the growing season characteristics of the main meteorological factors in the grassland ecosystem from April to September of 2018. Wind speed (WS) changed relatively smoothly, with a range of 0.01–4.48 m/s and an average of 2.42 m/s. The largest change occurred in April (Figure 7a). Relative humidity (RH) generally exhibited a slowly fluctuating downward trend, with smaller fluctuations from June to August. Meanwhile, the water vapor mass in the saturated air at the same temperature and pressure remained constant or increased, resulting in a slowly fluctuating downward trend in RH (Figure 7b). Photosynthetic active radiation (PAR) exhibited relatively minor variations in June, July, and August compared to other months (Figure 7c). The saturation water vapor pressure difference (VPD) increased with rising temperatures in June, July, and August (Figure 7d). In June, July, and August, both air temperature (Ta) and soil temperature (Ts) were higher, while they were lowest in April. The air temperature (Ta) overall showed a normal distribution, with mean values of 10.85 °C and

10.86 °C at 2 m and 10 m, respectively. The soil temperature at 0 cm varied more, while the soil temperature at 20 cm varied less, and the soil temperature was similar in all strata (Figure 7e,f). Soil moisture content (SWC) varied significantly among the strata, with the highest value at 5 cm, the lowest at 20 cm, and the second-highest at 10 cm, with mean values of 0.22 m$^3$/m$^3$, 0.18 m$^3$/m$^3$, and 0.12 m$^3$/m$^3$, respectively (Figure 7g). Soil heat flux (SHF) showed a slowly decreasing trend overall, with significant changes in April and May, and then leveled off in the remaining months (Figure 7h).

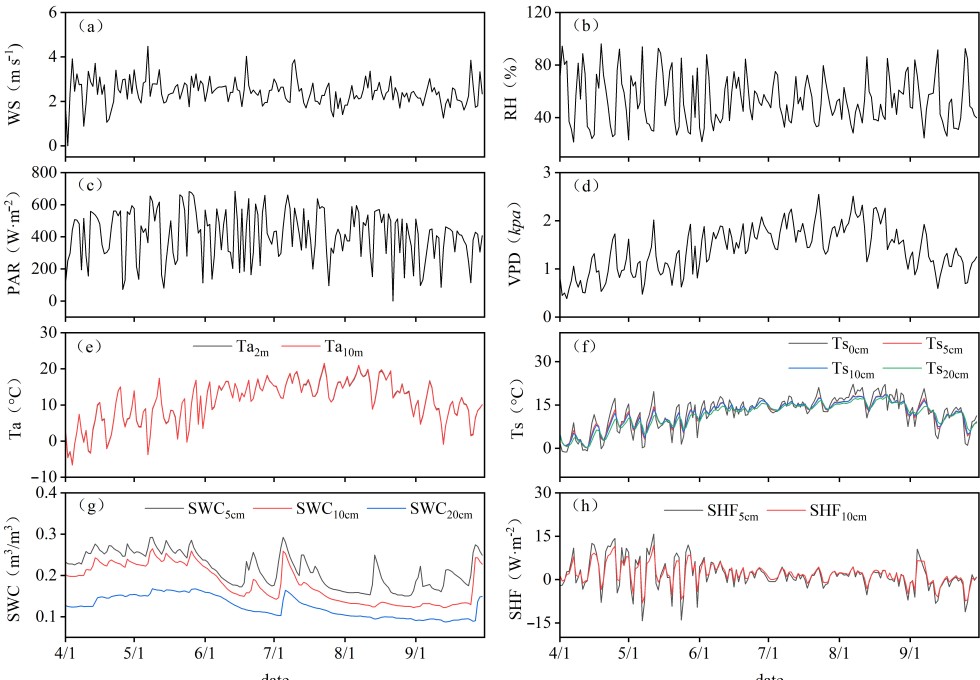

**Figure 7.** Seasonal variation in meteorological factors in grassland ecosystems from Apr to Sept of 2018. (**a**) Seasonal variation in wind speed (WS); (**b**) seasonal variation in relative humidity (RH); (**c**) seasonal variation in photosynthetic active radiation (PAR); (**d**) seasonal variation in vapor pressure difference (VPD); (**e**) seasonal variation in air temperature (Ta); (**f**) seasonal variation in soil temperature (Ts); (**g**) seasonal variation in soil moisture content (SWC); and (**h**) seasonal variation in Soil heat flux (SHF).

### 3.2.2. Contribution of Meteorological Factors to $CO_2$ Fluxes

The machine learning algorithm, random forest (RF), has the advantage of being able to handle large amounts of mixed data with high noise immunity and assess the importance of each variable factor [43]. Therefore, the contribution of meteorological factors to $CO_2$ fluxes at half-hourly and daily scales was calculated using the RF model.

As shown in Figure 8, at the half-hourly scale, the 0 cm soil temperature had the greatest contribution to NEE and GPP, while the 20 cm soil temperature had the largest contribution to Reco. It is noteworthy that for the studied seasonal variations and control factors of water and $CO_2$ flux in alpine meadows in Lijiang, southwestern China, using eddy covariance technology on half-hourly and daily scales, photosynthetically active radiation (PAR) and air temperature are the primary meteorological factors determining net ecosystem production (NEP) [44]. At the daily scale, the 20 cm soil moisture contributed the most to NEE and GPP, followed by the 10 cm and 20 cm soil temperature, and the 5 cm soil temperature contributed the most to Reco. This indicates that soil temperature is a prerequisite for the variation in $CO_2$ fluxes in the Middle Tien Shan grassland ecosystem because temperature affects the enzyme activity of plant physiological processes, which in turn affects photosynthesis in the ecosystem. It also suggests that the meteorological factor

of soil temperature has the most significant influence on $CO_2$ fluxes, both at the half-hourly and daily scales.

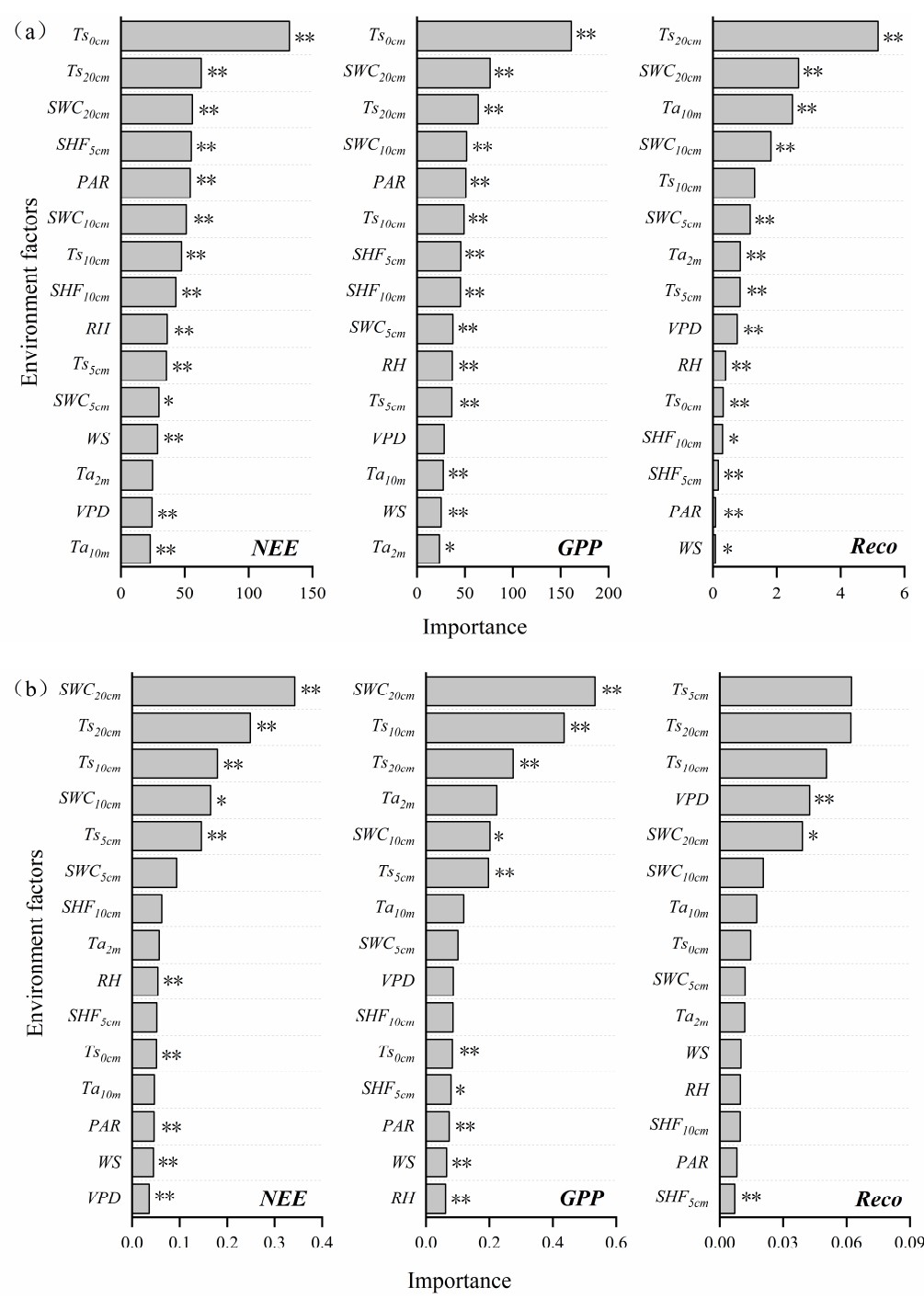

**Figure 8.** Ranking of the contribution of meteorological factors affecting $CO_2$ fluxes. (**a**) On a half-hourly scale; and (**b**) on a daily scale. ** represents passing the 99% significance test; * represents passing the 95% significance test.

## 3.3. Response of $CO_2$ Fluxes to Temperature

### 3.3.1. Relationship between $CO_2$ Fluxes and Temperature

To further characterize the effect of soil temperature on $CO_2$ fluxes, we fitted the function curves of soil temperature and $CO_2$ fluxes for each stratum (Figure 9). The study showed that soil temperature and $CO_2$ fluxes at different depths exhibited a univariate

linear regression, and the correlation between soil temperature and $CO_2$ fluxes improved with the increasing soil temperature. Soil temperature and NEE at different depths exhibited a negative correlation, meaning that NEE decreased as the soil temperature increased. The correlations between $Ts_{0cm}$, $Ts_{5cm}$, $Ts_{10cm}$, and $Ts_{20cm}$ with NEE are 0.04, 0.09, 0.10, and 0.11, respectively (Figure 9a–d). The correlation between soil temperature and GPP at different depths was positive, indicating that the respiration of the grassland ecosystem increased more than photosynthesis with the increase in soil temperature, leading to increased GPP. The correlations between $Ts_{0cm}$, $Ts_{5cm}$, $Ts_{10cm}$, and $Ts_{20cm}$ with GPP are 0.16, 0.25, 0.26, and 0.27, respectively (Figure 9e–h). The relationship between soil temperature at different depths and NEE and GPP was tested for significance, effectively demonstrating that soil temperature is an important indicator influencing carbon flux in the Middle Tien Shan grassland ecosystem.

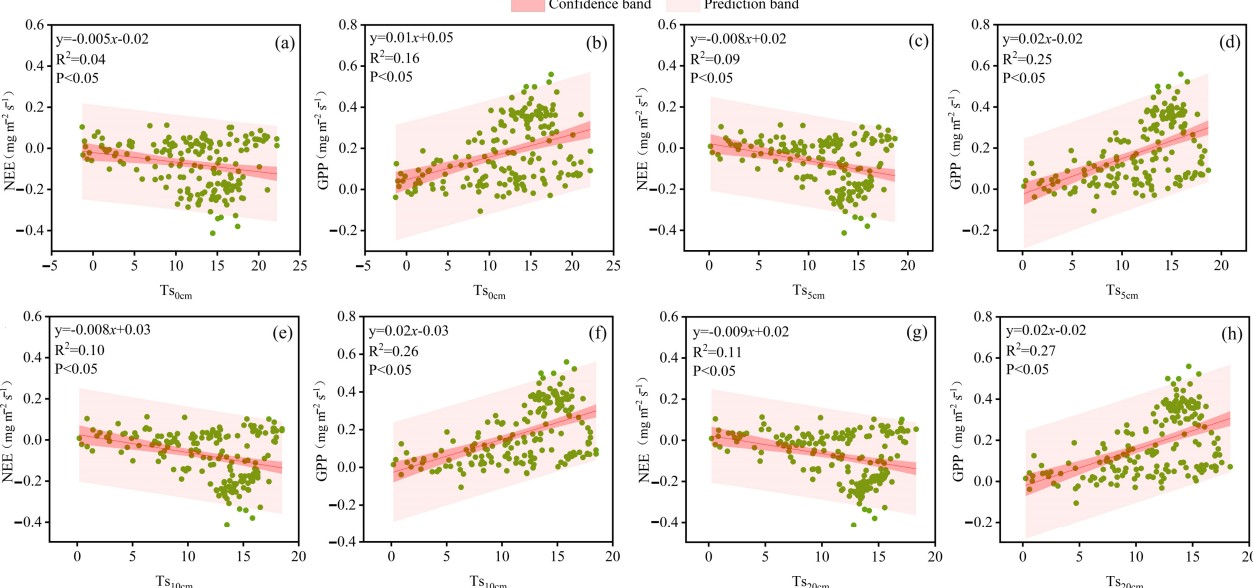

**Figure 9.** Relationship between $CO_2$ fluxes and soil temperature at different depths. (**a**) The relationship between 0 cm soil temperature and NEE, (**b**) the relationship between 0 cm soil temperature and GPP, (**c**) the relationship between 5 cm soil temperature and NEE, (**d**) the relationship between 5 cm soil temperature and GPP, (**e**) the relationship between 10 cm soil temperature and NEE, (**f**) the relationship between 10 cm soil temperature and GPP, (**g**) the relationship between 20 cm soil temperature and NEE, and (**h**) the relationship between 20 cm soil temperature and GPP.

To further investigate the effect of temperature on carbon fluxes in grassland ecosystems, a curve was fitted as a function of temperature and $CO_2$ fluxes using different temperature gradients (Figure 10). From the figure, it can be observed that, similarly to the relationship between soil temperature and $CO_2$ fluxes, the temperature of different gradients showed a univariate linear regression with a negative correlation between NEE and temperature, and an exponential distribution with a positive correlation between GPP and temperature. The correlation between air temperature and $CO_2$ fluxes was significantly higher than that of soil temperature, indicating that air temperature is also an important factor in regulating $CO_2$ fluxes in grassland ecosystems. However, the contribution of air temperature to $CO_2$ fluxes was significantly lower than that of soil temperature (Figure 8).

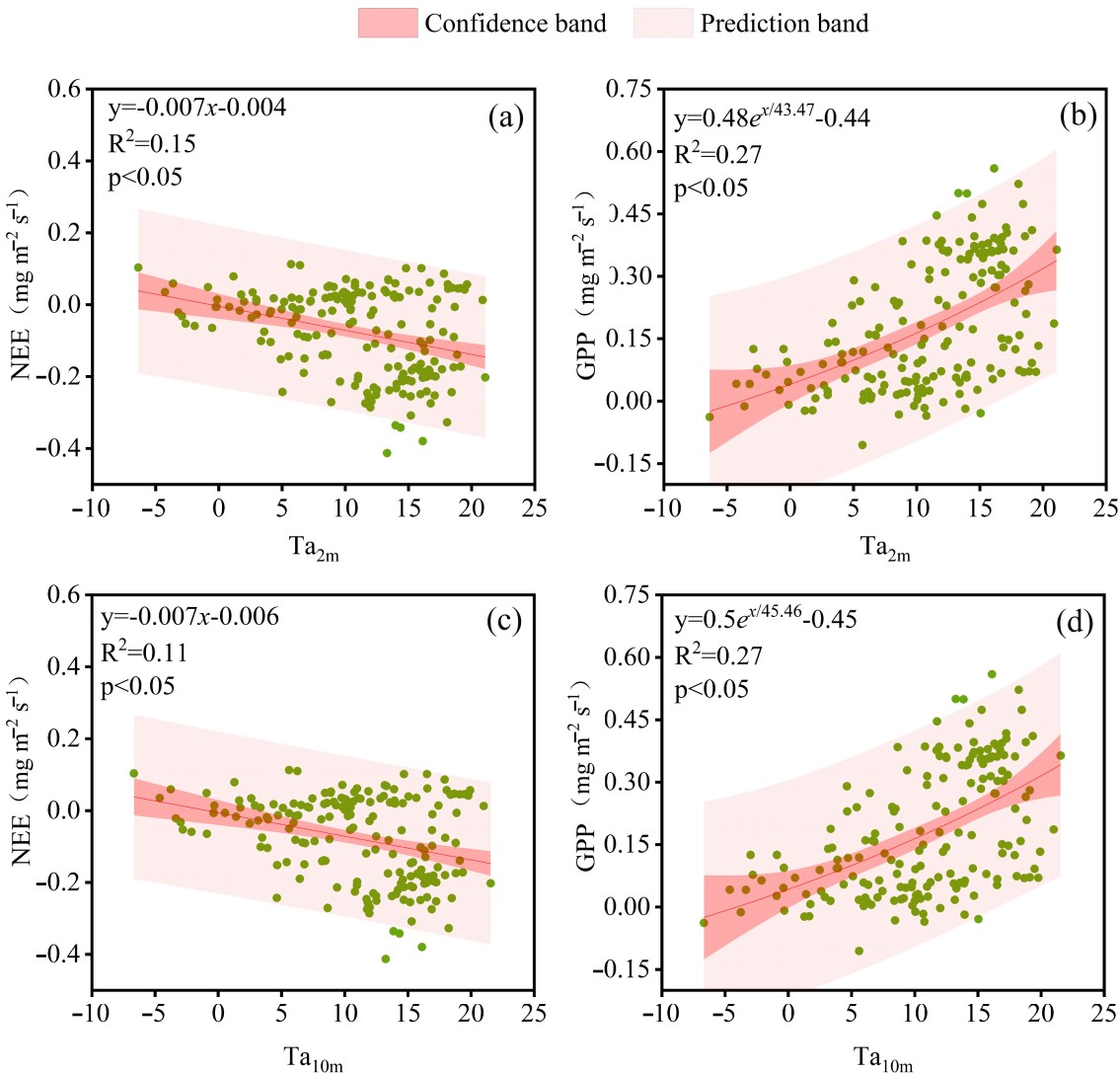

**Figure 10.** Relationship between $CO_2$ fluxes and air temperature with different gradients. (**a**) The relationship between 2 m air temperature and NEE, (**b**) the relationship between 2 m air temperature and GPP, (**c**) the relationship between 10 m air temperature and NEE, and (**d**) the relationship between 10 m air temperature and GPP.

### 3.3.2. Critical Values of Temperature Effects on $CO_2$ Fluxes

The soil and air temperatures were divided into eight and six intervals, respectively, to investigate $CO_2$ fluxes in different temperature ranges (Figure 11). The study showed that as the temperature increased, NEE tended to decrease and then increase, with a "turning point" at 25 °C (Figure 11a). When the soil temperature reaches 25 °C, it provides an optimal temperature for vegetation growth, and the stomata of vegetation roots open to efficiently absorb photosynthetic radiation and increase the photosynthetic absorption rate [16]. At this point, photosynthesis is greater than respiration in grassland ecosystems, leading to a relatively low concentration of $CO_2$ in the air and causing the grassland to generally act as a carbon sink. When the soil temperature drops to below 15 °C, vegetation is vulnerable to cold stress, and prolonged low temperatures are detrimental to crop growth and development, leading to reduced photosynthesis in the grassland ecosystem. This, in turn, weakens photosynthesis, causing $CO_2$ fluxes in the air to begin rebounding (Figure 11b). The trend of GPP is opposite to that of NEE, and the critical soil temperature for GPP is 25 °C (Figure 11c). The critical soil temperature for the effect on Reco is 10 °C.

Reco begins to increase when the soil temperature exceeds 10 °C and reaches its maximum value when the soil temperature is above 40 °C (Figure 11e).

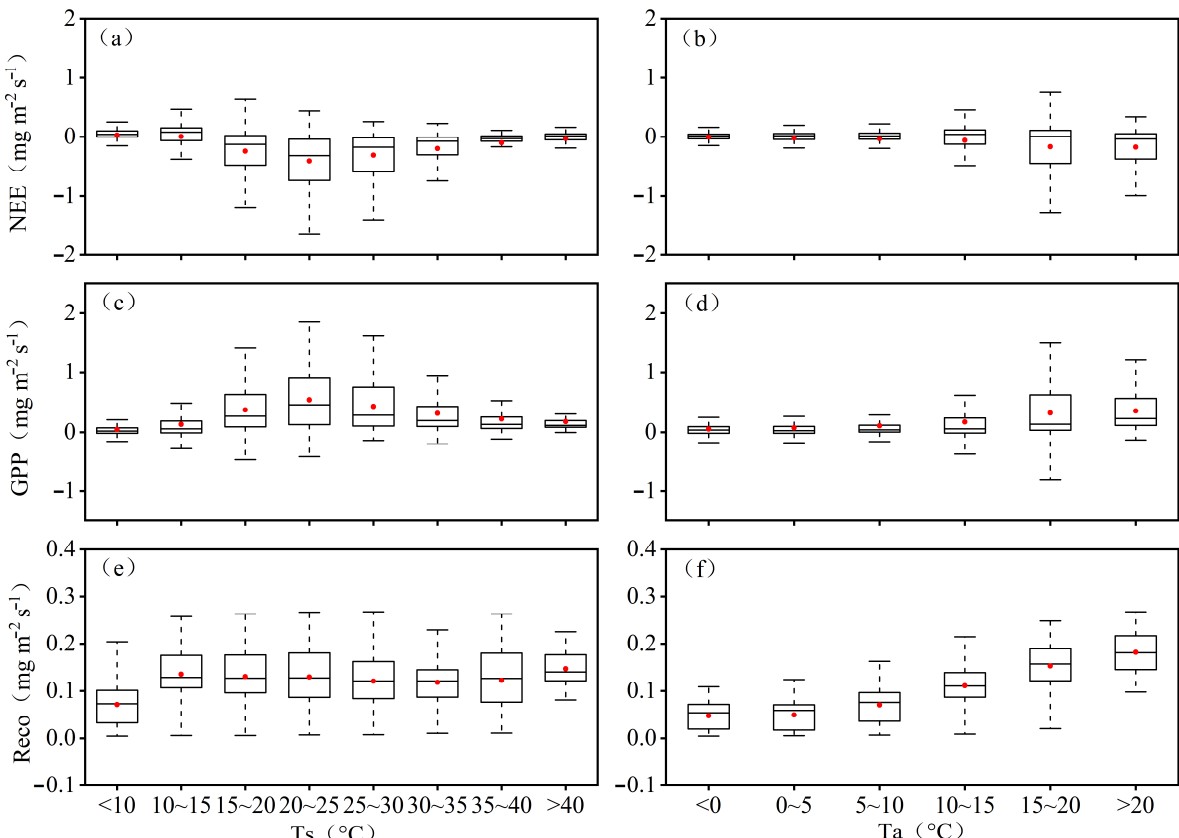

**Figure 11.** Critical values of the effect of soil temperature and air temperature on $CO_2$ flux. (**a**,**c**,**e**) Critical values of the effect of soil temperature on $CO_2$ flux, and (**b**,**d**,**f**) critical values of the effect of air temperature on $CO_2$ flux.

NEE decreases gradually with increasing air temperature, and changes more smoothly when the air temperature is below 15 °C. However, it decreases more rapidly when the air temperature exceeds 15 °C (Figure 11b). Conversely, the trend of GPP is opposite to that of NEE, with GPP showing an increasing trend when the temperature exceeds 15 °C (Figure 11d). Nonetheless, Reco increases with temperature and does not demonstrate a critical value due to its unique geographical location with relatively low temperatures.

### 3.4. The Trend of Carbon Flux at the Regional Scale

#### 3.4.1. Assessment of the Applicability of GCFD Data

Figure 12 shows a comparison between the field-measured values of carbon flux and the GCFD data during the 2018 growing season in the Tian Shan grassland. The field-measured values of NEE, GPP, and Reco show a high correlation with the GCFD data. Compared to GPP and Reco, the correlation between the field-measured values of NEE and the GCFD data is relatively low at 0.59, while the correlation between the field-measured values of GPP and Reco and the GCFD data is 0.81 and 0.73, respectively. The *RMSEs* are 5.21, 4.99, and 2.20, and the *Bias* values are 0.41, 0.19, and 0.94, respectively (Figure 12a–c). The variations in the field-measured values and the GCFD data are generally in phase. The field-measured values of NEE and the GCFD data show a reversed "U"-shape trend, reaching their minimum values in June and July. However, the field-measured values have a larger magnitude of variation compared to the GCFD data. The field-measured values of GPP and Reco exhibit the same trend as the GCFD data. Starting from April, both the field-measured values and the GCFD data show an increasing trend, reaching their peak

values in July, followed by a decreasing trend. However, the field-measured values have a larger magnitude of variation compared to the GCFD data. The trend of Reco in the field-measured data is similar to that of GPP, with both reaching their maximum values in July. However, the GCFD data overall tend to overestimate compared to the field-measured values (Figure 12d–f).

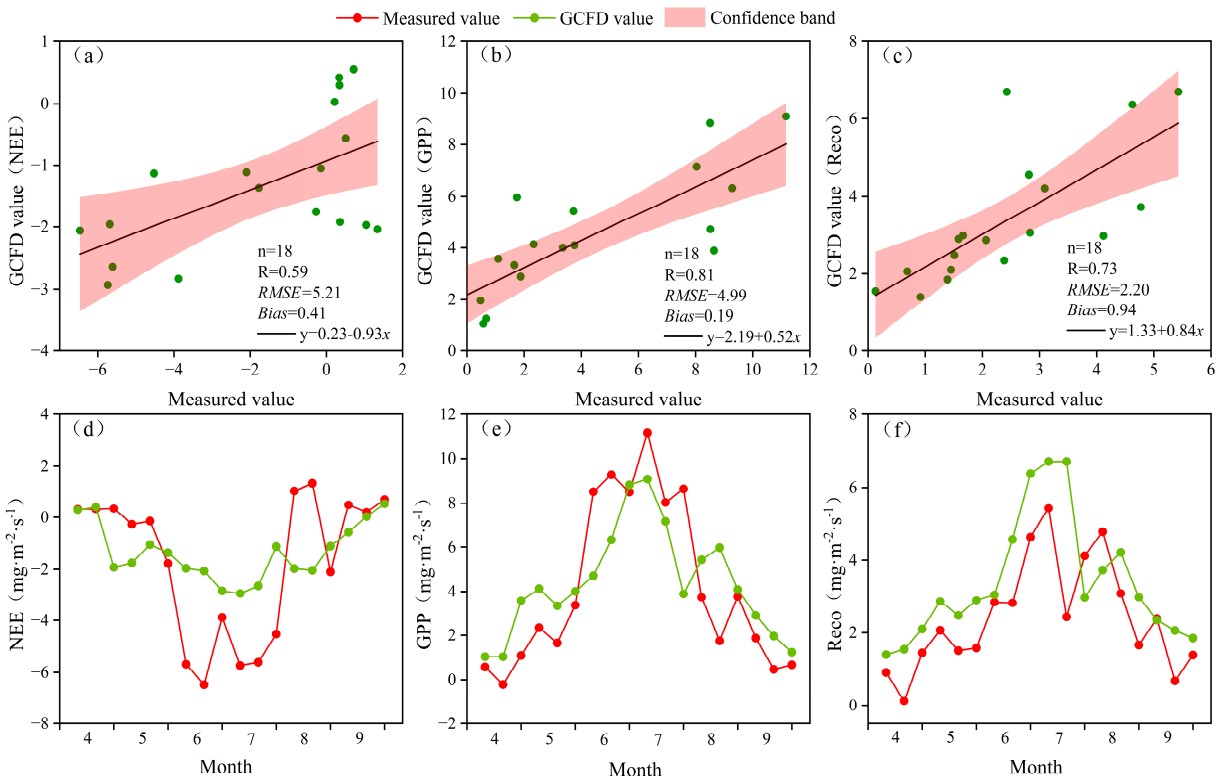

**Figure 12.** Comparison and validation of the GCFD data and the measured value at the Ulastai Station from April to September of 2018. (**a**–**c**) The correlation and error between GCFD data and measured value of $CO_2$ flux data, and (**d**–**f**) comparison of the trend changes between GCFD data and measured value of $CO_2$ flux data.

### 3.4.2. Remote Sensing Distribution of the GCFD Dataset

Figure 13 shows a remote sensing distribution map of the average carbon flux during the growing season in the Tien Shan region of Xinjiang in 2018. The minimum overall change in NEE occurs in the Ili River Valley, which is $-5.91$ mg m$^{-2}$ s$^{-1}$. This is because the Ili River Valley has a significantly warmer and moister climate compared to other regions from April to September, leading to vigorous vegetation growth. During this time, photosynthesis exceeds respiration, resulting in a lower $CO_2$ flux in the air and the occurrence of a carbon sink phenomenon. In contrast, in the eastern and western regions of Tien Shan, where there are more high mountains and glaciers, the sparse vegetation leads to a less pronounced carbon sink phenomenon. Ulastai station shows a more significant carbon sink phenomenon, with an average NEE value of $-1.48$ mg m$^{-2}$ s$^{-1}$ during the growing season. This is due to the thriving grassland ecosystem in the Ulastai station from April to September. The overall changes in GPP and Reco in the Tien Shan region exhibit an opposite trend to NEE, with the maximum values occurring in the Ili River Valley at 11.58 mg m$^{-2}$ s$^{-1}$ and 12.35 mg m$^{-2}$ s$^{-1}$, respectively. The average values in the Ulas Plateau during the growing season are 4.44 mg m$^{-2}$ s$^{-1}$ for GPP and 3.22 mg m$^{-2}$ s$^{-1}$ for Reco.

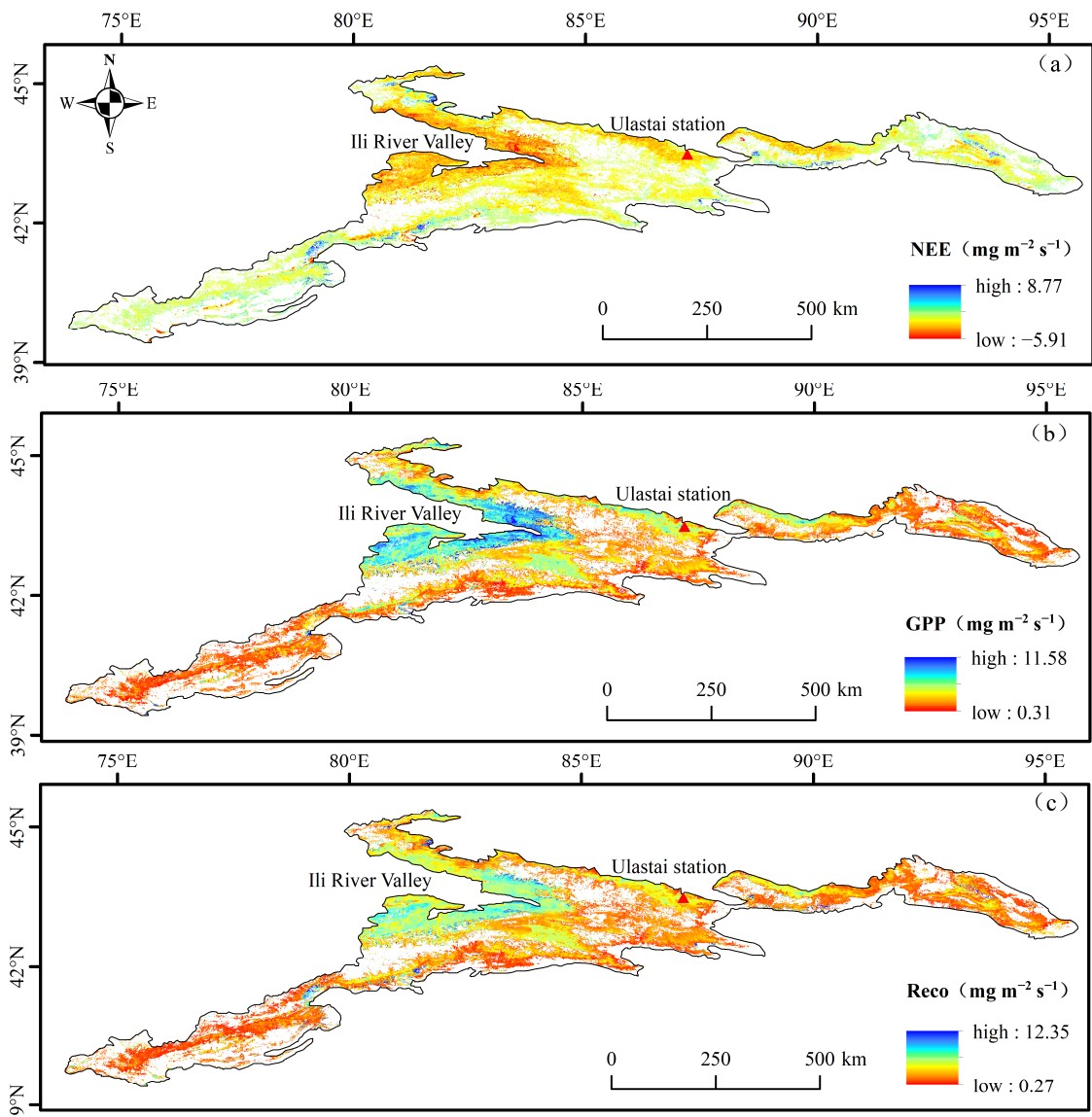

**Figure 13.** Average seasonal distribution of carbon flux in the Tien Shan region from April to September of 2018. (**a**) NEE, (**b**) GPP, and (**c**) Reco.

## 4. Discussion

### 4.1. Variations in $CO_2$ Fluxes

The NEE of the Middle Tien Shan grassland ecosystem showed an overall "U" curve change from April to September 2018. The ecosystem acted as a net carbon sink during the day and a net carbon source during the night. At 5:00, NEE showed a decreasing trend, while GPP showed an increasing trend due to the increase in solar height and radiation, leading to greater levels of photosynthesis than respiration in the grassland ecosystems [45]. This conclusion is consistent with the findings of previous research conducted by Du et al. in wetlands [18]. In terms of daily variation, NEE and GPP showed opposite trends, with the maximum $CO_2$ uptake occurring on June 17 at $-9.73$ g C m$^{-2}$ d$^{-1}$. Similar findings were reported in a study of a rice ecosystem in the Khorqin grassland; the difference is that Bao et al. found that in the rice ecosystem, the maximum $CO_2$ uptake occurs on August 15th, with a value of $-17.89$ g C$^{-2}$ d$^{-1}$ [19]. In terms of monthly variation, the overall cumulative $CO_2$ flux in the growing season of the Middle Tien Shan grassland ecosystem from April to September was $-329.49$ g C m$^{-2}$, 779.04 g C m$^{-2}$, and 449.55 g C m$^{-2}$, respectively. Compared to the reported $CO_2$ flux of $-183.45$ g C m$^{-2}$ in the growing season of the desert

ecosystem in the Gurbantunggut Desert by Gulnur et al. [46], this indicates that the carbon sequestration capacity of grassland ecosystems in the arid region of northwest China is higher than that of desert ecosystems.

Compared to other terrestrial ecosystems (Table 2), the Middle Tien Shan grassland ecosystem has a weaker carbon sink capacity than the forest [47] and meadow-rice ecosystems [19], but a stronger capacity than marsh and Siberian bog ecosystem [18,48]. The hydrothermal conditions play a crucial role in affecting the strength of carbon sequestration [49]. The sandy grassland of Horqin has a substratum dominated by fine sand and clay-powder grains that strongly absorb solar radiation, resulting in an increased potential evapotranspiration, and combined with less precipitation; this leads to a stronger carbon sequestration capacity of the Middle Tien Shan grassland ecosystem compared to the sandy grassland ecosystem of Horqin [50]. In comparison to the grassland ecosystem of the Yunnan–Guizhou plateau [12], the carbon sequestration capacity of the Middle Tien Shan grassland ecosystem is weaker. The Yunnan–Guizhou plateau belongs to the subtropical climate zone with abundant water and heat conditions, while the Middle Tien Shan grasslands belong to the typical temperate continental climate with less precipitation and a dry climate. This weaker capacity of carbon sequestration in the Middle Tien Shan grassland ecosystem is due to the lack of water and heat conditions.

**Table 2.** Comparison of $CO_2$ fluxes in different types of terrestrial ecosystems (values marked with * indicate that the study time scale is one year).

| Type | Study Period | Position | $CO_2$ (g C m$^{-2}$) | Reference |
|---|---|---|---|---|
| Grassland | April 2018~September 2018 | 43°28′N, 87°12′E | −329.49 | This study |
| Grassland | July 2017~August 2018 | 27°46′N, 107°28′E | −425.14 * | [12] |
| Sandy grassland | May 2015~September 2015 | 42°55′N, 12°42′E | −120.54 | [50] |
| Siberian bog | May 2015~August 2015 | 60.90°N, 68.70°E | −202 | [48] |
| Floating blanket marsh | January 2016~December 2016 | 25°07′N, 98°33′E | −233.8 * | [18] |
| Forest | January 2015~December 2015 | 30°06′N, 78°12′E | −526.87 * | [47] |
| Meadow—rice | May 2020~October 2020 | 43°20′N, 122°37′E | −769.24 | [19] |

*4.2. The Relationship between $CO_2$ Fluxes and Meteorological Factors and Their Response to Temperature*

The RF model was used to calculate the correlation between $CO_2$ fluxes and meteorological factors in the Middle Tien Shan grassland ecosystem, and it was found that soil temperature was the main meteorological factor affecting $CO_2$ fluxes. As the soil temperature increased, the NEE of the Middle Tien Shan grassland ecosystem decreased, while GPP and Reco increased. This study is consistent with the previous conclusion that GPP and Reco increase significantly with global warming [51]. The carbon sink during the growing season occurs in June and July, which is consistent with favorable water and thermal conditions, despite the highest temperature occurring in August. This result may be attributed to the extreme drought climate in August, indicating that water and thermal conditions are important factors limiting photosynthesis in arid vegetation [52]. In August, grassland ecosystems were susceptible to high-temperature stress due to enhanced solar height and solar radiation, which, together with low precipitation, accelerated the shortening of the grassland phenological cycle, leading to increased potential evapotranspiration and respiration in the grasslands. Enhanced evapotranspiration could lead to water stress in plants [53], resulting in weaker $CO_2$ flux uptake and release in grassland ecosystems, as well as weaker GPP and Reco. Previous studies have shown that drought or high temperatures cause the $CO_2$ balance of arid and semi-arid ecosystems to shift from carbon sink to carbon source [54–57].

The grassland ecosystem had the strongest carbon sequestration capacity when the soil temperature is 25 °C. Above this temperature, the grassland ecosystem is prone to heat stress, while below this temperature, it is susceptible to cold stress. The current investigation confirms this conclusion, finding that when soil temperature rises above 35 °C, extreme

drought conditions are highly likely to occur. Drought and high temperatures lead to dehydration of grass cells, and the lack of water and heat conditions adversely affect the photosynthesis of grassland [58]. When the soil temperature exceeds 10 °C, the Reco of grassland ecosystems begins to increase. As the soil temperature surpasses 40 °C, Reco reaches its maximum value. Air temperature also had a strong correlation with carbon fluxes in the grassland ecosystems, which was consistent with previous studies [59–61]. In our research, we have discovered that the temperature threshold for the occurrence of carbon sinks in grassland ecosystems is 15 °C. This further demonstrates that vegetation requires the most suitable temperature for growth through photosynthesis [62].

*4.3. Remote Sensing Carbon Flux*

Through a comparison and analysis of field measurement data from the Ulastai Station in the Middle Tien Shan grassland ecosystem, it was found that the GCFD data are applicable to the Ulastai region. The correlation coefficients between GCFD data and the measured values of NEE, GPP, and Reco are 0.59, 0.81, and 0.73, respectively. The accuracy of GPP and Reco in the GCFD dataset was higher than NEE, which is consistent with the conclusions reached by Shangguan et al. [33]. The *RMSE* values for the GCFD data compared to the measured values of NEE, GPP, and Reco are 5.21, 4.99, and 2.20, respectively, the *Bias* values are 0.41, 0.19, and 0.94. The reason for this error may be due to the fact that the training samples were selected from 280 global sites, mainly distributed in Europe and North America. Additionally, there may be regions with a lack of observational data, resulting in inaccurate simulated accuracy and errors between the site measurements and the GCFD dataset. Furthermore, there is an imbalance in the temporal and spatial resolutions of the remote sensing data, meteorological data, and carbon flux data used in the GCFD dataset. To unify the spatiotemporal resolution, the time resolution for these three types of data is set to 10 days per step, and the spatial resolution is set to 1 km. This process may introduce deviations in the dataset results [63,64].

Figure 12 presents the remote sensing spatial distribution of carbon flux in the Tien Shan region of Xinjiang. The Ulastai area shows a significant carbon sink from April to September 2018, with the highest carbon sink value occurring in the Ili River Valley, which is consistent with previous studies [65]. Therefore, through the verification analysis of field measurement data from the eddy correlation system and GCFD data, it was shown that the errors in the GCFD dataset resulting from the uneven distribution of training sample sites and unified spatiotemporal resolution are not significant for evaluating global carbon cycling. The example of Ulastai Station provides a scientific basis for the application of GCFD data in other regions. It also verifies the feasibility of using a machine learning fusion algorithm to construct a carbon flux dataset and provide data support for areas with sparse measurements.

**5. Conclusions**

The focus of this study was on the characteristics of carbon dioxide fluxes during the growing season and their response to temperature in the grassland ecosystems of the Middle Tien Shan Mountains in Xinjiang. The grassland ecosystems acted as carbon sinks during the daytime from 6:00 to 18:00 and as carbon sources during the rest of the day. Due to the increase in solar altitude and solar radiation, the carbon dioxide flux of the grassland ecosystem changed most in June, July and August, with June and July showing significant carbon sinks. The soil temperature and air temperature at different depths were negatively correlated with NEE and positively correlated with GPP and Reco. The carbon sequestration capacity of the grassland ecosystems was strongest when the soil temperature was 25 °C.

The GCFD data were compared and analyzed with the data from the Ulastai station. They show a high correlation and small errors compared to the measured value. This dataset is highly applicable in the Ulastai region and has consistent remote sensing spatial distribution. The GCFD dataset can clearly demonstrate the characteristic changes in

carbon flux in the Tien Shan region, providing possibilities for the future application of GCFD data in other areas.

This study only used the 2018 growing season eddy covariance data to explore the carbon flux balance. The carbon flux variation characteristics of grassland ecosystems during the non-growing season is worth further investigations. Additionally, this study only utilized site-specific data to reveal the influencing mechanisms of carbon flux in grassland ecosystems, which implies the need for further exploration of the influencing mechanisms of carbon flux in remote sensing space.

**Author Contributions:** Conceptualization, K.Z. and A.M.; methodology, K.Z. and Y.L.; software, Y.W.; validation, C.W. and M.S.; formal analysis, F.Y., C.Z. and W.H.; data curation, J.G. and A.A.; resources, A.M.; writing—original draft preparation, K.Z.; writing—review and editing, K.Z., A.M. and Y.L.; visualization, K.Z. All authors have read and agreed to the published version of the manuscript.

**Funding:** This research was funded by the Special Project for the Construction of Innovation Environment in the Autonomous Region (PT2203), the Special Funds for Basic Scientific Research Business Expenses of Central-level Public Welfare Scientific Research Institutes (IDM2021005), the Scientific and Technological Innovation Team (Tien Shan Innovation Team) project (grant No. 2022TSYCTD0007), the National Natural Science Foundation of China (grant No. 41875023), the S&T Development Fund of IDM (KJF202302), the Special Funds for Basic Scientific Research Business Expenses of Central-level Public Welfare Scientific Research Institutes (IDM2021001), and the Graduate Education Innovation Program of the Autonomous Region (XJ2023G032).

**Data Availability Statement:** The data used in this paper can be obtained from A.M. (ali@idm.cn) upon request.

**Acknowledgments:** The carbon flux data of the eddy covariance system were provided by the Urumqi Desert Meteorological Research Institute of the China Meteorological Administration. The GCFD dataset was provided by the National Tibetan Plateau Scientific Data Center.

**Conflicts of Interest:** All authors declare no conflict of interest.

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
