# Peer review of "Temporal and Spatial Variations in Carbon Flux and Their Influencing Mechanisms on the Middle Tien Shan Region Grassland Ecosystem, China"

_remotesensing, doi:10.3390/rs15164091_

Round 1

Reviewer 1 Report

The manuscript “Temporal and spatial variations of carbon flux and their influencing mechanisms of the Middle TienShan region grassland ecosystem, China” by Kun Zhang et al. deal with carbon balance of grasslands in China. Authors present new data on net ecosystem exchange, gross primary production and ecosystem respiration for mountain grassland in 2018.

The manuscript theme is actual, the aim is clear. The introduction is full and describes the studied problem. Figures are colorful and support the text. In general, the manuscript is well written and can be published after revision.

The Methods section should be expanded by adding details about processing of GCFD dataset. I did not understand why section 3.4.4 is titled “Remote Sensing carbon flux”. Is it a spatial distribution of fluxes from GCFD dataset? If yes than it is a model data described by Shangguan et al.

Section 3.3 is not directly related to the manuscript task. It is discussed response of CO2 fluxes to temperature. Linear and nonlinear relations of NEE, GPP and Reco with soil and air temperature are described. While Reco were calculated directly from air temperature (see REddyProc). I recommend to remove this section from the paper.

Please expand the Discussion by details about the biases between observations and GCFD dataset. What are the possible reasons of the discrepancy? Are the biases significant for the regional carbon balance estimation? Add a conclusion about applicability of the GCFD dataset.

Specific comments

Line 16-17. The first sentence is not clear, please rephrase.

Line 49-50. What is the difference between alpine and mountain grassland? If it is the same ecosystem, please use a single word, if not explain the peculiarities.

Line 52 . hm2  - ? convert to km2

Line 129 – vortex correlation -> eddy –covariance

Line 132 – open-circuit -> open-path

Line 132 – Authors claim than 3D ultrasonic anemometer Wind Master Pro was used. Figure 3c demonstrate sonic anemometer CSAT3 from Campbell Scientific. What anemometer was used for observations? https://gillinstruments.com/wp-content/uploads/2022/08/WindMaster-Pro-iss-8.pdf or https://www.campbellsci.cc/csat3

Line 135 - The same question for net radiation sensor. 4-channel net radiometer NR01 mentioned in the text. Figure 3d demonstrate net radiometer CNR01 from Kipp&Zonen.

Line 134 – Figure 3d demonstrate 2D wind speed and wind direction sensor but not 3D ultrasonic anemometers. Please Replace Figure 3 or replace the text described the used equipment.

Line 190 – Expand the methods section by mentioning the applied method of NEE partitioning. You described both day and nighttime partitioning (splitting) methods. What method you used in this paper?

Line 227 – Please comment on positive GPP at nighttime. What are the reason of photosynthesis without solar irradiation?

Line 239 – Figure 4a – vertical axis should be CO2 flux or something similar.

Line 294 – I suggest to move Table 1 and its description into Discussion section.

Section 3.2.2 – Figure 7 should be explained. What is on the horizontal scale? How  the Importance of meteorological factors were calculated?

Line 420 – Please comment on how the data corresponded to the observation site were extracted from GCFD dataset. Is it was a single grid cell or interpolated from adjusted grid cells? Did you youse 1-km or 9-km version of GCFD?

Reviewer 2 Report

The current investigation entitled “Temporal and spatial variations of carbon flux and their influencing mechanisms of the Middle TienShan region grassland ecosystem, China” authored by Zhang et al., need substantial improvement.

Abstract:

The authors need to revise line 34-36, starting with “In nutshell ……” since the meaning is not clear at all. So, revise the whole statement.

Introduction section:

The introduction section needs significant improvement, since in the current state it looks like just the assemblage of the statement without any connection to the preceding and succeeding statement. So kindly revise the introduction section using the transition words such as “However, although, moreover, simultaneously, furthermore” for connecting the statement. For instance, in line 47, the authors outbox shifted to the research perspective, without providing any background of the research in the grassland. Moreover, from line 60-67, the statements are not clear, what author want to say. Is there only one study related to that grassland in this particular region.

Form Line 88-98. The author need to provide the novelty of the current investigation, previous research gap and specific objective or hypothesis of the current investigation in detail. The detail about the data utilization, methodology and procedure should be moved to the material and methods section.

Material and Methods

In figure 1. Provide the detail about the figure a, b and c in the caption. Moreover, the information about elevation, and use should be provided in the text also.  Line 163 the detail of the satellite images should bee provided as a table in the main manuscript or as supplementary file. I suggest authors to add one methodology framework or flowchart of methodology used in the present investigation.

Results: In the result section, authors have mentioned that the there is significant variation. However, author need to mention, the level of significance in the parenthesis. Figure 5, authors need to mention, what is the difference between thee seasonal variation and  monthly variation?? As, only growing season were taken into consideration. Moreover, in the result section, authors need to only provide the result and discussion should be provided in the designated discussion section. Thus, line 285-293, line 416, line 393-395, 402, 375-376along with other mentions should be moved to the discussion section. Similarly, the Table 1 should be part of the discussion section rather than result section.  Furthermore, author need to revise the word seasonal variation to monthly variation, since only one season i.e., growing season, so there is no seasonal variation.

The discussion section needs to elaborated with the comparison of recent findings to the present study finding . Moreover, the causes provided for different results founded in the current investigation should be supported with relevant citation/reference.

Conclusion

A constructive conclusion should be provided rather than just a replica of results.  Moreover, statement in line 555-557 should be removed. Moreover, at the end, the limitation of the current investigation with future research in this field should be provided.

Specific comment

Line 17. Kindly revise it as: Therefore it is important to study the carbon……..

Line 34. Revise  it as follow: In nutshell, remote sensing  spatial

Line 40. Should be revised as follow: The Carbon dioxide (CO2) fluxes in terrestrial ecosystems ……..

Line 44-46 need to be revised. Since the meaning is not clear.

Line 51-52. What is hm2?????

Line 55-56. The diversity and  species are the same thing. If not kindly justify.

Line 58-59. Does there any relevant citation?? Mention it.

Line 63. What is EC system. Mention it.

Line 64-67. There are various grasslands? What are there distribution??

TienShan or Tienshan. Kindly make it identical throughout the manuscript.

Line 105-107, the meaning is not clear. Kindly re-write.

Line 155. The meaning/full form of NEE, GPP and Reco need to mentioned.

Line 158. the dataset, need to mention GCFD. Moreover, the authors need to full form of the GCFD for better clarity to the readers.

Line 178, author should avoid to use statements like “In this paper” instead of this should use  “in current investigation”

Line 219. Instead of writing directly “Figure 4” the author should write like that “ The result from current  investigation indicated that the diurnal variation……”

 Extensive editing of English language required

Round 2

Reviewer 1 Report

Authors carefully addressed all isues from the first review. The manuscript can be published in the present form.

Reviewer 2 Report

Second Revision

The manuscript entitled “Temporal and spatial variations in carbon flux and their influ- 2 encing mechanisms on the Middle Tien Shan region grassland 3 ecosystem, China” authored by Zhang et al.,.

The author has made subsequent revision during the first round of the revision. Especially the introduction section is improved in well manner. However, before to be considered for the publication the author needs to taken care off following improvement. At the end of the introduction section author should provide one major application of the current investigation.

The statement in line 35-37, is somewhat creates misperception and thus should be revised.

In figure 3, authors need to provide the detail about the figures in a, b, c and so on for better clarity to the audience.

In the serial, Figure 9 is missing. Moreover, the figure 8-13, the author need to provide the caption of a, b,c ….

The result section is written well and worth publication, but the discussion section still lack more comprehensive and specific statement from the previous investigation while the general statements should be removed.

Line 600-604, in the conclusion section does not make any sense and thus should be removed from the conclusion, while author need to provide the limitations of the current investigation  

Minor editing of English language required
